# Patterns and Predictors of Low-Calorie Sweetener Consumption during Pregnancy: Findings from a National Survey

**DOI:** 10.3390/nu15194196

**Published:** 2023-09-28

**Authors:** Bereket Gebremichael, Zohra S. Lassi, Mumtaz Begum, Murthy Mittinty, Shao-Jia Zhou

**Affiliations:** 1Department of Food and Nutrition, School of Agriculture, Food and Wine, The University of Adelaide, Glen Osmond, Adelaide, SA 5064, Australia; bereketgebremichael.menota@adelaide.edu.au; 2Robinson Research Institute, The University of Adelaide, Adelaide, SA 5006, Australia; zohra.lassi@adelaide.edu.au (Z.S.L.); mumtaz.begum@adelaide.edu.au (M.B.); 3College of Health Science, Addis Ababa University, Addis Ababa AA 1000, Ethiopia; 4School of Public Health, Faculty of Health and Medical Sciences, The University of Adelaide, Adelaide, SA 5000, Australia; 5Adelaide Medical School, The University of Adelaide, Adelaide, SA 5000, Australia; 6College of Medicine and Public Health, Flinders University, Adelaide, SA 5042, Australia; murthy.mittinty@flinders.edu.au

**Keywords:** artificial sweetener, non-nutritive sweetener, pregnancy, consumption pattern, latent class analysis, Australia

## Abstract

Recently, the World Health Organization recommended avoiding low-calorie sweeteners (LCS) during pregnancy due to concerns that it may be linked to adverse pregnancy outcomes and offspring wellbeing. This study examined the patterns and predictors of LCS consumption among pregnant women in Australia. A survey was conducted among 422 pregnant women aged 18–50 years. Sociodemographic, lifestyle, dietary intake including LCS consumption, pregnancy-related characteristics, and awareness about the health effects of LCS were assessed. We used latent class analysis and multinomial logistic regression to identify LCS consumption patterns and predictors of consumption patterns, respectively. The mean (SD) age of the women was 30 (4.6) years. Three LCS consumption patterns were identified: infrequent or non-consumers representing 50% of the women, moderate consumers encompassing 40% of the women, and the remaining were habitual consumers. Over two-thirds (71%) of women were not aware of the potential adverse effects of LCS, and only a quarter of them were concerned about the possible adverse effects on their health and their offspring. Increasing age and living with a medical condition decreased the likelihood of moderate consumption by 7% and 55%, respectively. Frequent sugar-sweetened beverage consumption and gestational diabetes predicted habitual LCS consumption. This research suggested widespread LCS consumption among pregnant women in Australia, but lower awareness of its potential adverse health effects. Interventions to increase awareness of potential adverse effects are warranted.

## 1. Introduction

Excessive sugar intake has been identified as one of the major drivers of the global obesity pandemic [1,2]. Low-Calorie Sweeteners (LCS), also called artificial sweeteners, have been promoted as a healthy alternative to sugar [3]. Consequently, consumption of LCS has increased significantly over recent decades [4]. Despite the significant shift from consuming sugar-sweetened food and drink to LCS-containing alternatives, obesity prevalence, associated health complications, and comorbidities are rising globally [5].

LCS are food additives that offer the sensory property of a sweet taste like sugar but provide low or no calories [6]. LCS have long been considered to be metabolically inert, however, this view has recently been scrutinised due to concerns regarding their potential adverse health effects [7,8]. Evidence from systematic reviews and meta-analyses regarding LCS consumption and the risk of chronic diseases in humans is inconsistent. Reviews of observational studies showed that LCS consumption is associated with adverse health outcomes, such as increased risk of obesity, diabetes mellitus, and cardiovascular problems [9,10], while reviews of randomized and non-randomized clinical trials reported no associations [11,12,13]. Maternal consumption of LCS in pregnancy has also been linked with increased birth weight, preterm delivery, and offspring obesity later in life in a systematic review of observational studies [14]. One of the proposed mechanisms of action is that LCS may interfere with the regulation of glucose and energy homeostasis. The intense sweetness coupled with low or zero energy in LCS has been shown to trigger a neuronal starvation response and promote food intake [15]. LCS consumption might elicit a comparable response to sugar intake, potentially resulting in increased glucose absorption and reduced insulin sensitivity [7]. LCS affect the gut microbiota composition, leading to impaired glucose tolerance [16].

Although the causal effect of in utero exposure to LCS and the long-term health of the offspring has not been established in humans, concerns have been voiced regarding the safety of LCS consumption during pregnancy [17,18]. In animal models, LCS consumption during pregnancy altered sweet taste preference and metabolic dysregulations in the offspring (e.g., increased risk of obesity and microbiome dysbiosis) later in life [19].

Given the ubiquity of LCS in processed foods and beverages [20] and their potential adverse impact on offspring, it is important to assess how common LCS consumption is among pregnant women and what drives their consumption. Studies conducted in the USA between 2007–2014 and in Ireland between 2007 and 2011 reported that the prevalence of LCS consumption during pregnancy was 24% and 34%, respectively [21,22]. A limited number of studies reported the prevalence of LCS consumption among pregnant women, and none of the studies assessed women’s awareness regarding the potential adverse effects or patterns and predictors of LCS consumption during pregnancy. To our knowledge, how common LCS consumption among pregnant women in Australia has not been investigated. Although the recent WHO guidelines discourage the use of LCS during pregnancy [23], identifying subgroups with high consumption (frequent and from a variety of food sources) and what predicts different consumption patterns would help identify those at increased risk of potential adverse health outcomes and prioritise interventions. Therefore, this study aimed to (1) determine the proportion of LCS consumption in pregnant women in Australia and identify LCS consumption patterns, (2) explore predictors of LCS consumption patterns, (3) examine women’s awareness of the potential adverse health effects of LCS consumption.

## 2. Materials and Methods

### 2.1. Participants and Recruitment

We recruited pregnant women aged 18 to 50 years living in Australia between September and October 2022 through Qualtrics, a reputable and widely used panel provider for researchers to conduct online surveys. We included women with a good understanding of English and consented to participate in the study. The web-based survey was set up as an anonymous survey accessible via a URL link provided. To further ensure the eligibility of participants who agreed to participate in the study, we asked two screening questions at the beginning of the survey, including age and current pregnancy status. Based on their responses to the screening questions, if they were not pregnant at the time or outside of the eligible age range, they were ineligible to take part, and the survey was terminated. Otherwise, we applied no restrictions (see Appendix A).

### 2.2. Sample Size Estimation

Based on limited evidence, the prevalence of LCS consumption during pregnancy ranged from 24% [21] to 34% [22]. We used an average prevalence of 30% for the sample size estimation. To achieve 80% power and 95% precision, a minimum of 324 participants were required. There are no clear guidelines or consensus regarding the estimation of sample size for latent class analysis, however, the model-based technique and different traditional cluster analysis support the use of minimal power between 0.7 to 0.8 to correctly identify the number of classes [24]. Considering the above premises, we performed power analysis for the latent class model using two-tailed logistic regression formula in G*Power. A sample of 400 participants was required to achieve 70% power with an expected relative risk ratio (RRR) of 1.3 and moderate association (0.50) to the latent class and an error of 0.05. Hence, we considered the current sample (*n* = 422) adequate to identify consumption patterns and their predictors.

### 2.3. Assessments

The survey questionnaire was informed by literature review [21,22,25,26], which covered the following aspects: (1) sociodemographic (age, education attainment, employment status, country of birth, state of residence, and postcode), (2) lifestyle and pre-pregnancy related characteristics (pre-pregnancy weight and height, pre-pregnancy and pregnancy alcohol consumption, smoking, physical activity, supplement use, and general medical conditions), (3) pregnancy-related characteristics (parity, gestational age at the time of survey, and pregnancy complications), (4) dietary intake including LCS consumption, reasons for consuming LCS, and (5) safety concerns about the consumption of LCS.

#### 2.3.1. LCS Consumption

We assessed LCS consumption from twelve food groups that are common sources of LCS based on previous studies [22,26,27] (see Appendix A). We asked respondents to report the frequency of consumption for each of the twelve food groups during their pregnancy using a six-point scale (Never, <1 time per week, once per week, 2–4 times per week, 4–6 times per week, and daily).

#### 2.3.2. Assessment of Covariates

We made a priori selection of covariates through literature review [21,28,29,30,31,32] and disciplinary knowledge. The covariates assessed included age, employment status, educational attainment, pre-pregnancy alcohol use, general medical conditions, pre-pregnancy BMI, Sugar-Sweetened Beverage (SSB) consumption, diet quality, moderate physical activity during pregnancy, parity, gestational age, pregnancy supplement use, pregnancy complications, and concerns about the potential adverse health impact of LCS.

Medical conditions were assessed by asking participants whether they had any existing medical conditions. If they responded affirmatively, they were subsequently asked to provide a list of those conditions. The responses were then categorised into three groups: “no” (indicating no medical conditions), “diabetes mellitus”, and “other”, which included any other medical conditions. We also asked about any complications during the current pregnancy. The responses were then classified into three categories: “no complications”, “gestational diabetes”, and “others”. To assess the level of moderate physical activity during pregnancy, we asked about the frequency of participation in moderately intense exercises throughout the current pregnancy. We provided a definition of moderate physical activity, describing it as any activity that requires some effort while still allowing the individual to engage in a conversation [33]. Responses were categorised into (≤2 times/week, 3–4 times/week, and ≥5 times/week). Safety concern about LCS was assessed by two questions: first, we asked if they were aware of any potential side effects of LCS, and the second question asked if participants were concerned that LCS may affect their health and the health of their offspring. Response options included “I don’t know”, “I am not sure”, and “Yes”. Additionally, we calculated pre-pregnancy BMI from self-reported pre-pregnancy weight and height.

We used the Index for Relative Socioeconomic Advantage and Disadvantage (IRSAD) developed by the Australian Bureau of Statistics [34] to examine the diversity of our sample in terms of socioeconomic advantage and disadvantage. The IRSAD index is a composite measure of socioeconomic position, including key indicators such as education, employment/income, housing, etc., at a local area level according to the postcode of an area. We extracted the IRSAD percentile score for each postcode from the Australian Bureau of Statistics [34]. Participants were assigned an IRSAD percentile score based on their postcode. A higher percentile score indicates a relatively higher level of advantage. We categorised the percentile score into quintiles.

Diet quality was assessed using a brief six-item food frequency questionnaire [25], which assessed the intake of the Five Food Groups defined in the Australian Dietary Guidelines (ADG), including vegetables, fruit, meat and alternatives, dairy, and grains [35]. We asked women to estimate the number of serves consumed from each of the Five Food Groups and ‘discretionary choices’ during an average week of their pregnancy. Examples of a variety of foods with an amount equivalent to one serving were provided for each food group [35]. Responses were recorded as the number of servings ‘per day’ or ‘per week’ or ‘per month’ or ‘never.’ We calculated the average number of daily servings consumed for each food group. Then we calculated the percentage of adherence to the ADG recommended intake for each food group as follows: *the number of daily serves consumed divided by the recommended number of daily serves × 100%*. The overall adherence to the ADG recommendation was calculated as the average adherence of the five food groups and used as an indicator of dietary quality. A high score for adherence indicated a better diet quality.

### 2.4. Statistical Analysis

To describe the characteristics of study participants, we report frequency and percentage for categorical variables, mean with SD for continuous data that visibly appeared to be normally distributed. To examine if LCS consumption patterns varied by participant characteristics, we used ANOVA for continuous variables and the Pearson *X*^2^ test for categorical variables. To calculate the proportion of any LCS consumption, we included all women who consumed at least one of the twelve food groups containing LCS regardless of the frequency or quantity.

#### 2.4.1. Latent Class Analysis

We used latent class analysis to identify LCS consumption patterns because the potential health risks associated with LCS consumption may differ depending on the food groups consumed, which may contain different types and amounts of LCS [36,37]. Latent class analysis is considered the most suitable method to take these factors into consideration and identify distinct subgroups of women based on their patterns of LCS consumption [38]. Latent class analysis, sometimes referred to as the mixture model [39], is a powerful statistical tool to identify subgroups of individuals (i.e., latent classes) within large, heterogeneous populations for categorical data. The model is useful for building typologies based on observed variables [40].

Latent classes were determined based on the consumption pattern from the twelve food groups used to measure LCS consumption. As recommended by Sinha et al., categories that have low frequency are difficult to fit into the latent class model because there is limited distributional information. In that case, collapsing the categories together is necessary [39]. Before fitting the latent class model, the twelve food groups measured using a six-point scale were recategorised depending on the commonness of their use. Three of the indicator variables, including soft drink, cordial drink, and lollies, were recoded into four categories as never, rarely (<1/week), sometimes (once/week), and often (≥2/week); five of the indicator variables, including yoghurt, ice cream, iced tea, biscuit, and LCS in cooking or baking were recoded into three categories as never, rarely (<1/week), and sometimes-to-often (≥1/week); the remaining four food groups, including pudding, energy drinks, protein drinks, and milk, were recoded into binary outcome as No (those who responded never) and Yes (indicating the combination of all responses except never). The graphical presentation of the relationship between class membership indicators, latent class and predictors of latent class is shown in Appendix A.

Latent class analysis was performed iteratively to identify models that provided the best fit to the data. We began with a parsimonious one-class model, then we fitted a series of models with an increasing number of classes from one to five classes. We assessed model fitness, which is the ability of a model to differentiate different consumption patterns, using likelihood ratio chi-squared statistic (*L2*), degrees of freedom (*df*), the Akaike Information Criterion (AIC), Bayesian Information Criterion (BIC), and Entropy *R^2^*. We selected the model with the best data-to-model fit based on the combined indicators of AIC, BIC, and entropy *R^2^*. Models with a lower AIC and BIC, and higher entropy *R^2^* indicate better models for differentiating consumption patterns [41,42]. Another consideration of model fit was the interpretability of each model and was assessed by whether the classes identified distinct groups (patterns). After determining the optimum number of classes, we then estimated the latent class probability and assigned participants to a class (assigned to the highest probability). Further, we determined the item response probability of each indicator variable for a standardised visual representation of identified classes.

#### 2.4.2. Predictors of LCS Consumption Patterns

After identifying the latent classes and assigning subjects to a class, we employed a hierarchical multinomial regression method to examine the relationship between latent classes and predictors. First, we modelled sociodemographic variables (age, employment status, and educational attainment) as Model 1. Model 2 included covariates in Model 1, and pre-pregnancy lifestyle-related variables (pre-pregnancy alcohol use, general medical condition, and pre-pregnancy BMI). Model 3 adjusted for covariates in Model 2 and additional pregnancy-related variables (supplement use during pregnancy, moderate physical activity during pregnancy, parity, pregnancy duration, pregnancy complications) and LCS safety concern, SSB consumption, and diet quality in pregnancy. Creating a hierarchical model in this fashion may lead to omitted variable bias. However, during pregnancy, women are often more attentive to their nutrition and make changes to their diets [43]. Therefore, we included pregnancy-related factors in the model as they might have a more pronounced impact on LCS consumption patterns compared to demographic and lifestyle factors. Using a hierarchical logistic regression model is appropriate for assessing predictors from these different sets of variables, allowing for an exploration of the incremental variance explained by each set of predictors [44]. We checked multicollinearity between covariates using the Variance Inflation Factor where values >2.5 indicated considerable collinearity [45]. We reported an adjusted relative risk ratio (aRRR) with a 95% confidence interval (CI). We used Stata version 17 (Stata Corporation, College Station, TX, USA) for analysis.

## 3. Results

### 3.1. Participant Characteristics

Overall, 422 pregnant women completed the survey. The mean age of women was 30.0 ± 4.6 years, and the mean BMI was 27.1 ± 6.5 kg/m^2^. Over three-quarters (81.7%) of women were working and 48.0% completed tertiary education (Table 1). Characteristics of participants across the three identified classes (consumption patterns) are also shown in Table 1. The habitual LCS consumers had a normal BMI and were highly educated (received tertiary education) and were living in socioeconomically advantaged areas. Women in the infrequent or non-consumer group were older and more likely not to be in the workforce than women in the other two classes. Other participant characteristics were similar in all three classes. Participant characteristics based on any consumption versus no consumption are shown in Appendix A. Mean (SD) daily servings from five food groups and the percentage of adherence to recommended daily intake for pregnancy are depicted in Appendix A.

### 3.2. Latent Class Identification

As shown in Table 2, the three-class model had a lower AIC and BIC, and higher entropy *R^2^*, indicating it is the best data-to-model fit.

### 3.3. Latent Class Description

Figure 1 shows class memberships of participants based on predicted latent class probability. Accordingly, 209 (49.5%) of the women were in Class 1, 169 (40.0%) were in Class 2, and 44 (10.4%) were in Class 3.

The item response probability of selected indicator variables of class membership is shown in Figure 2 and Figure 3. Class 1 was characterised by a low consumption of all foods and drinks containing LCS, with the probability <10% except for diet soft drinks (indicating 25% and 21% probability of consumption <1/week and ≥2/week, respectively). This class was labelled as “*infrequent or non-consumer*”. Class 2 labelled as “*moderate consumer.*” was characterised by moderate consumption of all food categories containing LCS. The probability of consumption ranged from 18% in LCS containing protein drinks ≥1/week to 51% consumption of diet soft drinks ≥2/week. Class 3 labelled as “*habitual consumer*” was characterised by a high probability of consuming all the food groups containing LCS as indicated with 72% probability of using LCS in cooking and baking ≥1/week and 90%, 80% and 70% probability of consuming LCS containing milk, protein drink and energy drink, respectively. The item response probability of selected indicator variables using a radar plot and line graph is shown in Appendix A.

### 3.4. Prevalence, Reason for Consumption, and Safety Perception

Our study showed that 95.0% [95% CI; 92.47–96.74] of the pregnant women reported consuming any LCS from at least one of the twelve food groups assessed. The main sources of LCS were diet soft drinks, chewing gums and lollies, biscuits, and cordial drinks, which were consumed by 51.4%, 37.2%, 31.3%, and 28.2% of women, respectively, at least once per week (Appendix A). The main reasons for LCS consumption reported were taste preference and trying to adopt a healthy lifestyle. Regarding awareness about the potential adverse effects of LCS, more than two-thirds of women were not aware of any potential adverse effect of LCS. Only about a quarter of women in all groups reported that they were concerned about the potential adverse effect of LCS on their health as well as their children. There were no substantial differences among women in the three LCS consumption groups in terms of general safety concerns and safety concerns about the foetus (Table 3).

### 3.5. Predictors of Moderate and Habitual LCS Consumption

The predictors of moderate and habitual LCS consumption patterns are shown in Table 4. Women who reported frequent consumption of SSB (≥2 times/week) or had gestational diabetes were approximately three times more likely to have a habitual LCS consumption pattern compared with women who consumed SSB less frequently [aRRR = 3.17, (95% CI; 1.39–7.21)] and without gestational diabetes [aRRR = 3.53, (95% CI; 1.03–12.10)]. In contrast, those with a medical condition were about 80% and 55% less likely to be habitual consumers and moderate consumers than those with no medical conditions, respectively. BMI also predicted habitual LCS consumption. Age was negatively associated, while supplement use and employment status positively predicted moderate LCS consumption. Other demographic, lifestyle-, and pregnancy-related factors showed mixed relationship with LCS consumption patterns. We observed no multicollinearity among covariates. See Appendix A for a full model output. Appendix A shows ordinal logistic regression model output based on the overall LCS consumption frequency.

## 4. Discussion

To our knowledge, this is the first study to report the proportion of pregnant women consuming LCS during pregnancy in Australia and the only study that assessed LCS consumption patterns and their predictors in pregnant women. The findings suggest that LCS consumption was widespread among pregnant women who participated in the study. We identified three distinct LCS consumption patterns: infrequent or non-consumers, moderate consumers, and habitual consumers. Further, we identified predictors of moderate and habitual LCS consumers compared with infrequent or non-consumers. Age, diet quality, and having a general medical condition were found to be negatively associated with moderate LCS consumption. Frequent SSB consumption (≥2 times/week) and having GDM predicted the likelihood of habitual LCS consumption. Only a quarter of the women were concerned about the potential adverse effect of LCS on their health as well as their offspring.

### 4.1. Prevalence of LCS Consumption

The high rate of LCS consumption among pregnant women is not surprising, given the prevalence of consuming LCS-containing foods and drinks in the general population was 66% two decades ago [26]. A recent report by the Australia Bureau of Statistics also showed that throughout 2019–2021, per capita diet beverage consumption per month averaged 10.5% more compared with the same month in the preceding year [46]. In addition, studies conducted in the US [21], Ireland [22], and Canada [47] also reported a relatively high prevalence of LCS consumption (ranging from 24–34%) during pregnancy. However, the proportion of women who reported consumption of any LCS in this study was higher compared with previous studies. The difference could be due to the fact that other studies assessed diet soft drink consumption as a proxy for LCS consumption [47]. This could have underestimated the prevalence in those studies since LCS are widely present in other food sources. Second, the previous studies used a single 24-h recall dietary assessment [21] or a three-day dietary record [22], which may not reflect habitual intake. Third, the other studies were conducted between 2003 and 2014. The consumption of LCS may have increased since then as more studies have focused on the potential adverse health impacts of LCS consumption in recent years [21]. Lastly, online surveys are prone to self-selection bias, and our sample may not be representative of the pregnant women in Australia.

### 4.2. Predictors of Moderate and Habitual LCS Consumption

The current study showed that over two-thirds of the women were unaware of any potential adverse effect of LCS, and only a quarter of the women were concerned about the possible adverse effect of LCS on their health as well as their offspring. To our knowledge, no published studies examined the awareness among pregnant women, but the awareness was similar among the general public in a Saudi Arabia study [48] and higher among the adult population in the United Kingdom [49]. Although not statistically significant, awareness about the potential adverse effects of LCS was negatively associated with moderate and habitual consumption patterns in this study. This is supported by a study conducted in the United Kingdom that found an association between awareness about adverse effects and reduced LCS consumption [49]. The scope of our assessment regarding awareness of potential adverse effects was limited. Further qualitative research to understand the reasons for the lower awareness or concern regarding the potential adverse health effects of LCS will offer valuable insights into women’s knowledge, attitude, and behaviour regarding LCS consumption during pregnancy.

A recent dose-response meta-analysis of 39 articles showed that an increase in the frequency or amount of LCS consumed was associated with an increased risk of adverse health outcomes [37]. Those who are frequently exposed to LCS and consume LCS from diverse food groups could be at an increased risk of potential adverse health effects. Similarly, those with moderate and habitual consumption identified in this study are likely to be at an increased risk of adverse health outcomes.

Our study showed that women with a medical condition were found to have lower likelihood of habitual LCS consumption patterns. A possible reason could be that those with medical conditions could be wary to consume LCS. However, further investigation is necessary to confirm the association. Our finding also show that age was negatively associated whereas education and employment were positively associated with LCS consumption. This is supported by studies in general populations [22,29]. A plausible reason for this could be that those who are younger are more likely to have frequent exposure to LCS marketing, and those with higher education could be more aware of the health effects of sugar consumption and are more likely to substitute LCS for sugar.

Women with GDM are recommended to follow a low glycemic index diet. This may involve replacing high glycemic index foods and drinks with low glycemic index alternatives containing LCS [50]. Therefore, pregnant women with GDM are more likely to consume LCS. This was also true in this study. The habitual LCS consumption likely reflected the substitution of LCS for sugar among women diagnosed with GDM.

The finding that LCS consumption was higher among those who frequently consumed SSB was unexpected as LCS are generally used as a sugar replacement. The observed association could partly be due to the possible effect of LCS on appetite regulation and taste perception, leading to craving for more sweets [51]. The compound health effect of LCS and SSB is likely to be high, given that both are reported to be associated with adverse pregnancy outcomes and offspring health [52,53].

Our finding also indicate that higher pre-pregnancy BMI was associated with a lower likelihood of being a habitual LCS consumer. This could be because those who habitually consumed LCS could be more conscious about weight and use these sweeteners as part of their weight management strategy. However, further investigation is necessary to delve deeper into the association.

### 4.3. Strengths and Limitations

Although online surveys are efficient and have higher completion rate, they are prone to self-selection bias. Online surveys may overrepresent those with internet access and higher educational status [54]. Likewise, our study sample consisted of a higher percentage of women who completed post-secondary and tertiary education compared with pregnant women in Australia. Despite this, women in our study were comparable to pregnant women in Australia in terms of age, employment condition, and diversity of their country of birth (Appendix A) [55]. Further, the IRSAD score for areas indicated that our sample included participants from diverse socio-economic backgrounds. Another limitation is that we did not assess the types and amount of LCS consumed. Since different types of LCS have different physiological and health effects, it would have been good to investigate that. However, it was beyond the scope of this study. Nonetheless, the frequency of LCS consumption from a wide range of food sources assessed in our study may provide some indication of the types and quantity of LCS consumed. Another limitation is that the LCS consumption and other nutritional data were self-reported and are prone to reporting bias. The smaller sample size also resulted in high standard errors and wide confidence intervals in some of the covariate and predictor variables. Finally, findings from the study may not be generalisable to all pregnant women in Australia due to the limited sample size.

Although our study has limitations, it has several strengths. We assessed LCS consumption from a wide range of foods and beverages that are common sources, which provided a better indication of how widely LCS were consumed among pregnant women. We also assessed perceived health risks of LCS consumption and its long-term effects on the foetus and whether this affected consumption patterns, which was not previously investigated. Further, the use of latent class analysis enabled us to identify subgroups of pregnant women with moderate and habitual LCS consumption patterns and what predicted those patterns. Identification of women with moderate and habitual consumption patterns will help to inform intervention priority.

## 5. Conclusions

LCS consumption may be widespread with low awareness of its potential adverse health effects among pregnant women in Australia. Public health interventions to increase the awareness of potential adverse effect of LCS consumption, particularly among pregnant women with moderate and habitual consumption, are warranted. Future research with representative population samples in the Pacific region and globally to present context-dependent burden and drivers of LCS consumption during this critical period and to investigate the impact of LCS consumption on maternal and offspring health is warranted.

## Figures and Tables

**Figure 1 nutrients-15-04196-f001:**
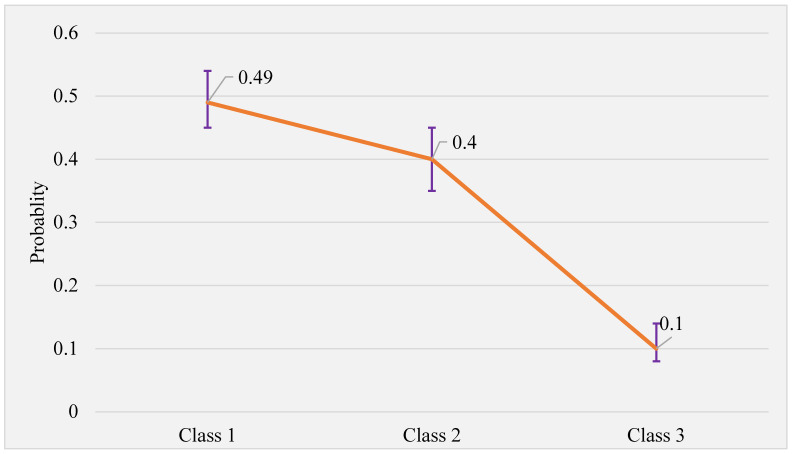
Predicted latent class probability with 95% *confidence interval*. Error bars indicate the 95% confidence interval.

**Figure 2 nutrients-15-04196-f002:**
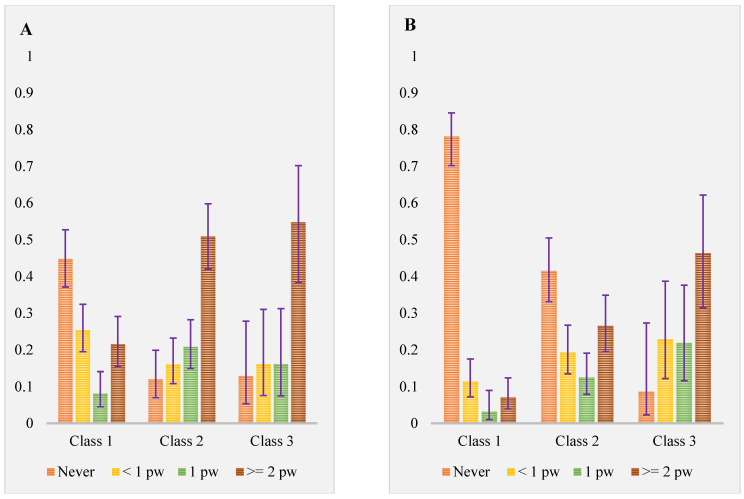
Estimated response probability of low-calorie sweetened food and drink consumption in each latent class (assessed as polytomous indicator variable for the latent class). (**A**) diet soft drink; (**B**) cordial drink; (**C**) ice cream; (**D**) LCS use in cooking and baking. *y*-axes indicate probability. Error bars indicate the 95% confidence interval.

**Figure 3 nutrients-15-04196-f003:**
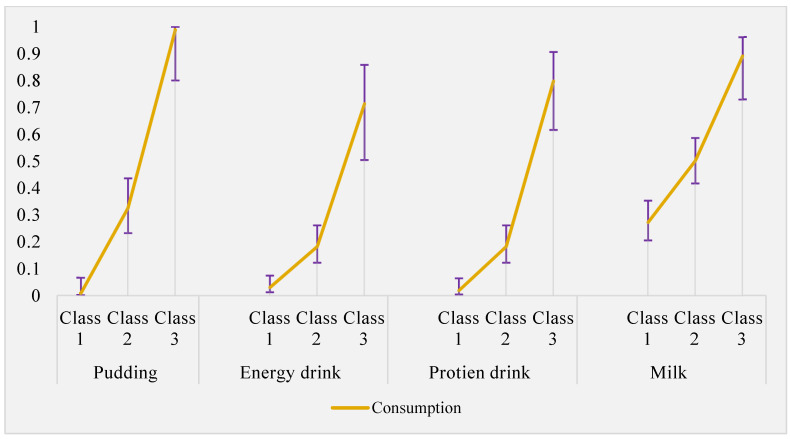
Estimated response probability of low-calorie sweetened food and drink consumption in each latent class (food groups assessed as dichotomous indicator for the latent class). Error bars indicate the 95% confidence interval.

**Table 1 nutrients-15-04196-t001:** Participant characteristics ^a^.

Characteristics	Category	Overall	Low-Calorie Sweetener Consumption Pattern			
			Infrequent or Non-Consumer (*n* = 209)	Moderate Consumer (*n* = 169)	Habitual Consumer (*n* = 44)	*p*-Value
Age (year)		30.0 ± 4.6	30.7 ± 4.3	29.5 ± 4.9	29.0 ± 4.8	0.013
BMI *(*kg/m^2^*)*		27.1 ± 6.5	27.3 ± 6.5	27.4 ± 6.6	24.8 ± 5.5	0.052
Diet quality score		47.2 ± 16.2	48.8 ± 15.7	45.4 ± 15.9	46.2 ± 18.6	0.124
Educational level	Secondary	82 (19.4)	41 (19.6)	34 (20.1)	7 (15.9)	0.037
	Post-secondary but no tertiary	148 (35.1)	65 (31.1)	72 (42.6)	11 (25.0)	
	Tertiary	192 (45.5)	103 (49.3)	63 (32.3)	26 (59.1)	
Employment status	Not working	77 (18.3)	50 (23.9)	22 (13.0)	5 (11.4)	0.011
	Working	345 (81.7)	159 (76.1)	147 (87.0)	39 (88.6)	
General medical condition	No	338 (80.1)	160 (76.5)	140 (82.8)	38 (86.4)	0.025
	Diabetes mellitus	17 (4.0)	5 (2.4)	9 (5.3)	3 (6.8)	
	Other	67 (15.9)	44 (21.1)	20 (11.8)	3 (6.8)	
Parity	Primigravida	169 (40.0)	79 (37.8)	71 (42.0)	19 (43.2)	0.387
	One	178(42.2)	98 (46.9)	64 (37.8)	16 (36.4)	
	Two or more	75 (17.8)	32 (15.3)	34 (20.1)	9 (20.4)	
Pregnancy stage	First trimester	81 (19.2)	37 (17.7)	32 (18.9)	12 (27.3)	0.419
	Second trimester	132 (31.3)	64 (30.6)	52 (30.8)	16 (36.4)	
	Third trimester	209 (49.5)	108 (51.7)	85 (50.3)	16 (36.4)	
Pregnancy complication	None	282 (66.8)	151 (72.2)	107 (63.3)	24 (54.5)	0.142
	GDM	44 (10.4)	18 (8.6)	19 (11.2)	7 (15.9)	
	Other	96 (22.7)	40 (19.1)	43 (25.4)	13 (29.5)	
Pre-pregnancy supplement use (Yes)	Yes	230 (54.5)	106 (50.7)	94 (55.6)	30 (68.2)	0.100
Pregnancy supplement use	Do not use	36 (8.5)	22 (10.5)	11 (6.5)	3 (6.8)	0.500
	Use but do not contain I and F	41 (9.7)	23 (11.0)	15 (8.9)	3 (6.8)	
	Use I and F containing supp	345 (81.7)	164 (78.5)	143 (84.6)	38 (86.4)	
Pre-pregnancy Alcohol use (Yes)	Yes	284 (67.3)	146 (69.8)	115 (68.0)	23 (52.3)	0.075
Pregnancy alcohol use (Yes)	Yes	28 (6.6)	9 (4.3)	14 (8.3)	5 (11.4)	0.125
Smoking habit	Never smoker	387 (91.7)	194 (92.8)	154 (84.6)	39 (88.6)	0.198
	Past smoker	29 (6.9)	10 (4.3)	14 (8.3)	5 (11.4)	
	Current smoker	6 (1.4)	5 (2.4)	1 (0.6)	-	
Moderate physical activity during the current pregnancy	≤2/week	324 (77.8)	165 (78.9)	131 (77.5)	28 (63.6)	0.178
	3–4/week	77 (18.2)	37 (17.1)	28 (16.6)	12 (27.3)	
	≥5/week	21 (5.0)	7 (3.3)	10 (5.9)	4 (9.1)	
IRSAD	First quintile (Disadvantaged)	78 (20.2)	38 (18.2)	32 (18.9)	8 (18.2)	0.027
	Second quintile	79 (20.5)	37 (17.7)	39 (23.1)	3 (6.8)	
	Third quintile	79 (20.5)	34 (16.3)	39 (23.1)	6 (13.6)	
	Fourth quintile	73 (18.9)	36 (17.2)	27 (15.9)	10 (22.7)	
	Fifth quintile (Advantaged)	77 (19.9)	38 (18.2)	23 (13.6)	16 (36.4)	

^a^ Data are Mean ± SD for continuous variables and frequency (%) for categorical variables. We used ANOVA to compare the mean between groups for continuous variables and chi^2^ test to compare proportion difference for categorical variables in relation to consumption patterns. BMI: body mass index (calculated from self-reported pre-pregnancy weight and heigh); GDM: gestational diabetes mellitus; I: iodine; F: folate; IRSAD: Index of Relative Socioeconomic Advantage and Disadvantage.

**Table 2 nutrients-15-04196-t002:** Fitness indicators of different latent class models.

Model	LL	*df*	AIC	BIC	L2	Entropy R2
1 Class	−4459.041	23	8964.083	9057.118	4148.479	-
2 Classes	−4069.316	47	8232.631	8422.746	3369.027	0.632
3 Classes	−3970.422	71	8082.844	8370.039	3171.240	0.716
4 Classes	−3931.431	91	8044.862	8412.958	3093.258	0.683
5 Classes	−3895.562	113	8017.125	8474.210	3021.520	0.687

AIC: Akaike Information Criteria; BIC: Bayesian Information Criteria; L2: likelihood ratio chi-squared statistic; LL: log likelihood ratio; *df*: degrees of freedom.

**Table 3 nutrients-15-04196-t003:** Reasons for low-calorie sweetener consumption and safety perception about LCS.

Variable	Category	LCS Consumption			
		Infrequent or Non-Consumer (*n* = 209)	Moderate Consumer (*n* = 169)	Habitual Consumer (*n* = 44)	*p*-Value
Are you aware of any adverse effects of LCS?	No	106 (50.7)	85 (50.3)	26 (59.1)	0.413
	Not sure	49 (23.4)	30 (17.7)	7 (15.9)	
	Yes	54 (25.8)	54 (31.9)	11 (25.0)	
Are you concerned that LCS may affect your or your baby’s health?	No	82 (39.2)	79 (46.7)	22 (50.0)	0.314
	Not sure	69 (33.0)	56 (33.1)	11 (25.0)	
	Yes	58 (27.8)	34 (20.1)	11 (25.0)	
Reason for consumption	Did not consume or no reason	116 (55.5)	33 (19.5)	4 (9.1)	<0.001
	Weight loss and/or DM	38 (18.2)	62 (36.7)	18 (40.9)	
	Other	55 (26.3)	74 (43.8)	22 (50.0)	

*p*-value is obtained from chi^2^ test. The infrequent or non-consumer group includes 21 participants reported they did not consume LCS. Other includes trying to adopt healthy lifestyle and test preference.

**Table 4 nutrients-15-04196-t004:** Multinomial logistic regression of factors predicting moderate and habitual low-calorie sweeteners consumption relative to infrequent or non-consumer (*n* = 416).

Variables	Category	Moderate Consumption		Habitual Consumption	
		aRRR ^a^	95% CI	aRRR ^a^	95% CI
Age	-	**0.93**	**0.88–0.98**	0.92	0.84–1.01
Employment status	Not working	Ref			
	Working	**3.25**	**1.67–6.33**	2.71	0.85–8.60
Educational level	Secondary	Ref			
	Above secondary but not tertiary	1.29	0.69–2.41	0.79	0.25–2.50
	Degree and above	0.77	0.41–1.45	1.37	0.46–4.07
General medical condition	No	Ref			
	Yes-DM	2.14	0.56–8.16	3.06	0.45–20.54
	Yes, other	**0.45**	**0.23–0.88**	**0.20**	**0.05–0.80**
Pre-pregnancy alcohol use	No	Ref			
	Yes	0.81	0.49–1.32	0.65	0.29–1.45
BMI	-	0.99	0.95–1.02	**0.89**	**0.83–0.97**
Diet quality	-	**0.98**	**0.97–0.99**	0.98	0.95–1.00
SSB consumption	≤once/week	Ref			
	≥2 times/week	1.34	0.82–2.19	**3.17**	**1.39–7.21**
Moderate physical activity during current pregnancy	No or <2/week	Ref			
	3–4/week	1.04	0.57–1.91	1.98	0.79–4.92
	≥5/week	1.74	0.58–5.18	3.92	0.93–16.51
Pregnancy supplement use	Not at all	Ref			
	Use, don’t contain I and F	2.29	0.76–6.85	4.07	0.54–30.29
	Use, contain I and F	**2.71**	**1.13–6.47**	3.77	0.80–17.74
Parity	Primigravida	Ref			
	One	0.90	0.54–1.49	0.89	0.37–2.13
	Two or more	1.93	0.96–3.89	3.06	0.94–9.93
Pregnancy duration	First trimester	Ref			
	Second trimester	0.96	0.49–1.87	0.85	0.30–2.38
	Third trimester	0.80	0.43–1.49	0.38	0.14–1.04
Pregnancy complication	None	Ref			
	Other	1.73	0.98–3.06	**3.11**	**1.26–7.67**
	GDM	1.27	0.55–2.95	**3.53**	**1.03–12.10**
LCS safety concern about the baby	No	Ref			
	Not sure	0.85	0.50–1.44	0.56	0.22–1.43
	Yes	0.61	0.34–1.10	0.57	0.22–1.51

^a^ Infrequent or non-consumer is the reference category. aRRR: adjusted Relative Risk Ratio; BMI: Body Mass Index; F: Folate; GDM: Gestational Diabetes Mellites; I: Iodine; LCS: Low Calorie Sweetener; SSB: Sugar Sweetened Beverage. Bold indicates significant association.

## Data Availability

Data pertaining to this article may be accessed by contacting the corresponding author with reasonable request.

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
