# Peer review of "Patterns and Predictors of Low-Calorie Sweetener Consumption during Pregnancy: Findings from a National Survey"

_nutrients, 2023, doi:10.3390/nu15194196_

Round 1

Reviewer 1 Report

This study conducted in Australia examined the consumption patterns of low-calorie sweeteners (LCS) among pregnant women. It identified three distinct LCS consumption patterns: infrequent or non-consumers, moderate consumers, and habitual consumers. Several factors were associated with these patterns, such as age, diet quality, having a general medical condition, frequent consumption of sugary beverages, and having gestational diabetes.

The study found that a significant proportion of pregnant women in Australia consume LCS, with moderate and habitual consumers making up a substantial portion of the surveyed population. Surprisingly, many women lacked awareness of the potential adverse health effects of LCS, and only a quarter expressed concern about these effects on their own health and that of their offspring.

Here are five concerns of the study that needs tto be assessed:

1. Self-Selection Bias: The study relied on an online survey, which may have led to self-selection bias. Participants who have internet access and are more tech-savvy might have been overrepresented in the sample. This could limit the generalizability of the findings to a broader population of pregnant women.

2. Lack of Detailed LCS Information: The study did not assess the types and specific amounts of LCS consumed by participants. Different types of LCS can have varying physiological and health effects, so not differentiating between them limits the depth of understanding regarding LCS consumption.

3. Single Data Collection Point: The study gathered data at a single point in time. This cross-sectional design doesn't allow for the examination of trends or changes in LCS consumption over time, and it can't establish causality between predictors and consumption patterns.

4. Sample Representation: While efforts were made to include participants from diverse socioeconomic backgrounds, the study sample consisted of a higher percentage of women with post-secondary and tertiary education compared to the general pregnant population in Australia. This could affect the generalizability of the findings.

5. Limited Awareness and Concern Assessment: The study assessed participants' awareness of potential adverse health effects of LCS consumption and their level of concern. However, the assessment was limited in scope and did not explore in-depth reasons behind the lack of awareness or concern, which could provide valuable insights into participant attitudes and behaviors.

6. There are several reviews that have come out in recent years highlighting the merits and shortcomings of LCDs. They should be included and discussed to improve the level of introduction and discussion. 

English is good

Reviewer 2 Report

This study examined the patterns and predictors of LCS consumption among pregnant women in Australia. Although the theme is relevant for Australia and science, this study has several limitations.

Abstract: To add more numbers and data from study.

Introduction: Unclear the hypothesis of study.

Methods: Sample size is small and did not reflect the Australia. In addition, this online survey is speculative and no details of regions and nutritional status os pregnancy are provided.

Results: Figure 3 is wrong (no appear). What is nutritional status? Body mass gain? biochemical parameters?

Discussion: No clear discussion regarding to regions in Pacific. 

Round 2

Reviewer 1 Report

The authors have improved the paper.

Strengths:

Comprehensive analysis of LCS consumption patterns among pregnant women, filling a gap in the literature.Use of latent class analysis to identify subgroups with different consumption patterns is a strong methodological choice. The paper addresses both the prevalence and predictors of LCS consumption, providing a well-rounded view of the issue.

Areas for Improvement:

The study relies solely on self-reported data, which is prone to reporting bias. Incorporating validated dietary assessment tools could enhance the reliability of the findings.

The paper could benefit from a qualitative component to explore the reasons behind low awareness or concern about LCS's potential health effects.

The study's limitations regarding sample size and representativeness should be more explicitly acknowledged, and the implications of these limitations for the study's generalizability should be discussed.

Discuss more review on sweeteners for instance:

10.3390/nu15173675

10.1017/S0007114523001484

10.1186/s43162-023-00232-1

10.3389/fped.2023.1200990

10.3390/nu15122711

10.21448/ijsm.1122618

10.5005/japi-11001-0029

10.17827/aktd.1039222

10.3389/fnut.2022.854074

10.56501/intjcommunitydent.v10i1.601

10.3390/nu14081682

10.1158/1940-6207.CAPR-21-0507

English is ok

Reviewer 2 Report

I think that manuscript has limited interest for readers of nutrients, but may be accepted for publication.

Author Response

There is no further comment from reviewer 2 that needs to be addressed.